# Evaluation of Health Risks Due to Heavy Metals in a Rural Population Exposed to Atoyac River Pollution in Puebla, Mexico

**Gabriela Pérez Castresana [1,2], Elsa Castañeda Roldán [3], Wendy A. García Suastegui [2]**, **José L. Morán Perales [2], Abel Cruz Montalvo [4] and Anabella Handal Silva [2,*]**

[1]  Posgrado en Ciencias Ambientales, Instituto de Ciencias, Benemérita Universidad Autónoma de Puebla, Puebla 72570, Mexico; perezcastresana@yahoo.es

[2]  Departamento de Biología y Toxicología de la Reproducción, Instituto de Ciencias, Benemérita Universidad Autónoma de Puebla, Puebla 72570, Mexico; wendy.garcias@correo.buap.mx (W.A.G.S.); joseluis.moran@correo.buap.mx (J.L.M.P.)

[3]  Centro de Investigaciones en Ciencias Microbiológicas, Departamento de Patogenicidad Microbiana, Instituto de Ciencias, Benemérita Universidad Autónoma de Puebla, Puebla 72570, Mexico; elsa.castaneda@correo.buap.mx

[4]  Departamento de Investigación en Ciencias Agrícolas, Instituto de Ciencias, Benemérita Universidad Autónoma de Puebla, Puebla 72570, Mexico; abcruz38@hotmail.com

*  Correspondence: anabella.handal@correo.buap.mx or ahandals@yahoo.com.mx; Tel.: +1-52-222-2295500 (ext. 7161)

**Abstract:** The health risks of Emilio Portes Gil's population, for the exposition to the Atoyac River pollution in the State of Puebla, was evaluated. The objective was to determine the concentration of nine heavy metals by ingesting water from wells and spri ngs. The chronic daily water intake (CDI), hazard quotient (HQ), hazard index (HI), and carcinogenic risk index (CRI) in adults, teenagers, and children were estimated. The results showed that the concentration of Fe, Al, Ni, and Pb in some of the samples exceeded the recommended standards for human consumption and was significantly higher in the dry season. The hazard index (HI), due to the collective intake of metals, was higher in children (>50% compared to adults), due to the consumption of spring water in the dry season. Risk of noncancerous diseases was not detected in the long term, since the indices did not exceed the unit (reference value). The carcinogenic risk from oral exposure to Cr ($CRI_{children} = 3.2 \times 10^{-4}$), was greater than the acceptable limit ($1 \times 10^{-6}$) in the water spring, and Cr and Pb were the main metals that contributed to the potential health risk of the inhabitants. The study showed the risks by the intake of polluted water from the sources of supply in the region, and that the risk is higher in the dry season (>100% compared with rainy season).

**Keywords:** water well; spring water; pollution water; risk evaluation; Atoyac river; E. Portes Gil

## 1. Introduction

The water crisis has an effect on the daily lives of the most disadvantaged or marginalized populations, who suffer from water-related diseases, living in degraded and often dangerous environments [1]. In Mexico, as in other developing countries, there are large inequalities in water supply and sanitation between urban and rural areas [2–5]; and 35 million Mexicans are in a situation of low availability [4,6]. Lack of access to a potable water system, or deficiency in supply through formal networks, has forced rural populations to look for alternative sources to cover their needs [2,5,7]. The use of groundwater through shallow wells has been a relatively low-cost and simple technology

alternative for these populations [1,7], particularly for those located in arid and semi-arid areas of the country (Central and Northern Mexico), and where rivers or other bodies of surface water are polluted [1,4,7]. An example of this is observed in the agricultural communities adjacent to the urban area of the city of Puebla (central Mexico), where the Atoyac River circulates, one of the most polluted in the country. In these towns, there are serious problems related to the supply of drinking water service and the water of the river cannot be used for domestic purposes because of its high degree of pollution, so they have to obtain water from the subsoil [8].

Despite the benefits provided by rural wells in terms of water availability, the wells are vulnerable to contamination, due to the lack of expert advice in the construction, protection, and depth thereof [7,9–11]; this is related to the fact that they have been drilled without government authorization or a concession title, and they are not registered in the Public Registry of Water Rights (REPDA, CONAGUA), so there is no control [12]. Knowledge about the diversity of chemical pollutants present in the water of the Atoyac River and its concentration is limited, but it is estimated that it must be high, considering that this river circulates through the fourth largest metropolitan area at national level (Metropolitan Area Puebla-Tlaxcala, ZMPT) and wastewater is discharged from hundreds of industries from different routes and the most populated municipalities, and in none of these cases are there tertiary treatment plants capable of eliminating specific pollutants, such as heavy metals [8,13–21]. These chemical elements, like other pollutants, are released as byproducts of both industrial processes and other human activities in the river water, and can be moved to more distant places, contaminating soils and other bodies of water [11–14].

The pollution of groundwater by heavy metals represents a threat to populations that drink water from these sources, due to the cytotoxic, carcinogenic, and mutagenic effects that many of these metals produce when they enter the human body, because although some are essential for human life, they are required in very low and specific concentrations [22–29]. These populations have a risk of developing diseases, such as central nervous system disorders, cardiovascular disorders, kidney failure, and osteoporosis, among others; in addition to predisposing children to deficient cognitive development and poor organ development [26,30,31]. In severe cases, heavy metals can increase the risk of developing cancer, as a result of the alteration in expression patterns of several genes [25,30,32–37]. Several scientific methodologies have been developed to evaluate the risk of cancerous and noncancerous diseases using equations based on the concentration of heavy metals in drinking water, and chronic exposure to them, taking into account the differences between groups of different ages, such as weight averages and the volume of drinking water consumed daily [38]. Thus, the hazard coefficient (HQ), the hazard index (HI), and the cancer risk index (CRI) have been used as a tool based on equations proposed by the United States Environmental Protection Agency (USEPA). 1986) [38–40].

Studies about the risk of disease in the Alto Atoyac region (Puebla-Tlaxcala), due to the exposure to heavy metals in groundwater do not exist; however, the risks due to exposure to heavy metals present in the soil of these regions or in the milk of cows fed with grass irrigated with wastewater have been quantified [41]. These studies show how the health of children is at high risk, due to the consumption of arsenic (carcinogenic element) present in the milk of cows, as well as how the men who work in agriculture are at risk of suffering diseases from dermal exposure to heavy metals present in soils [41,42]. There are no studies in the region about the pollution of rural wells and the risk to human health for groundwater intake.

The health is an important part of the quality of life, and for its conservation it is necessary to know which are the factors that can generate alterations, such as heavy metals or other chemical compounds, as well as to estimate the risk according to the degree of exposure to them [38]. The risk is defined by the USEPA [43] as the possibility of causing harmful effects to human health or ecological systems. The methodology used by this environmental protection agency allows characterizing the nature and magnitude of the risks due to chemical pollutants and other factors that may be present in the environment. The use of wastewater for irrigation of crops induces risks to the inhabitants of these

agricultural regions, as the pollutants are transferred to the soil, plants, groundwater, etc. [44,45] and the population is exposed to these elements by different routes. The exposure of heavy metals by the consumption of groundwater represents only one of the possible routes of exposure, it is important to consider that these rural populations obtain the water to drink mainly from the wells and springs and not from other sources [8].

The objective of this research was to determine the concentration of nine heavy metals in the wells and spring in the agricultural town of Emilio Portes Gil, affected by the pollution of the Atoyac River (Puebla, Mexico), using atomic absorption spectrophotometry, as well as determining the health risk to adults, adolescents and children of intaking groundwater, through the estimation of the chronic daily intake (CDI), hazard quotient (HQ), the total hazard index (HI)), and the cancer risk index (CRI) for each subgroup of the population, and its corresponding comparison between dry and rainy season.

## 2. Study Area

The study area, known as Emilio Portes Gil (EPG), belongs to the municipality of Ocoyucan, one of the 217 municipalities of the State of Puebla in Mexico. EPG is an ejidal colony of 522 inhabitants, whose main economic activity is agriculture [43,46] (Figure 1).

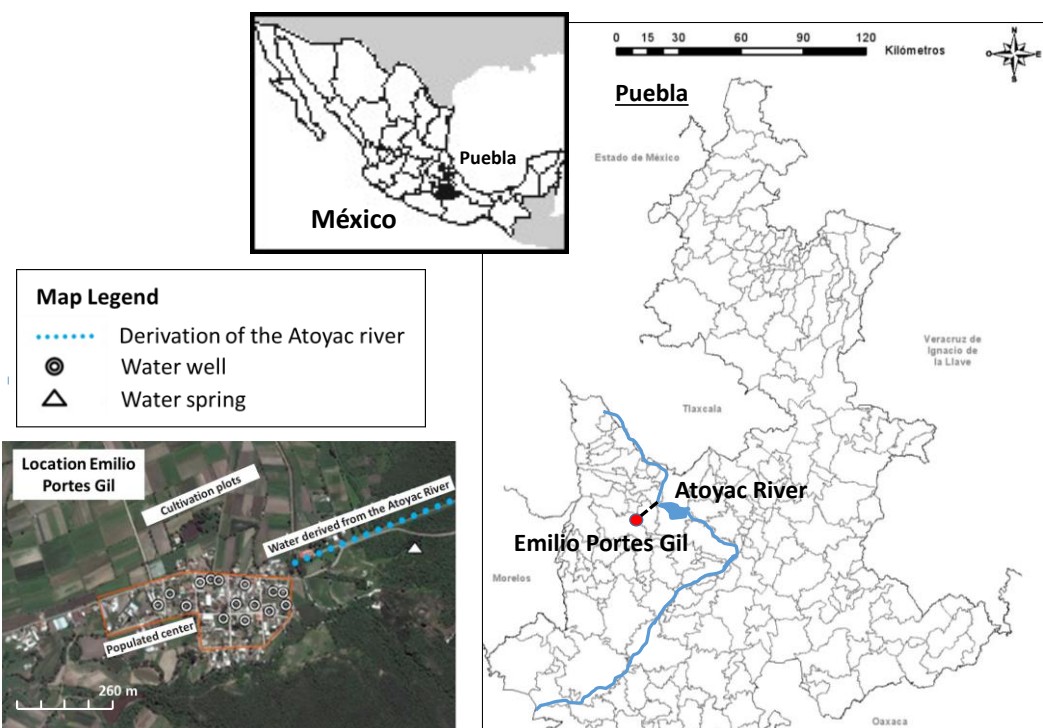

**Figure 1.** Study area, indicating the groundwater sources sampled in the agricultural town Emilio Portes Gil (Puebla, México).

It receives water from the Atoyac River through a channel covered in concrete, which has its origin in the Echeverria dam. The water that flows through the canal, before entering EPG, goes through the Portezuelo hydroelectric plant, where after generating electricity, it is incorporated into the main irrigation channel of the town (18°55′ 32.4″, N 098° 20′ 40.4″ O), to finally discharge into the Nexapa River [47]. Most homes (>90%) have tubing water that comes from a spring (1940 m above sea level) through a rudimentary pipe system. They receive the service approximately 3 days per week, which is why 70% of the houses have shallow wells (<20 m). As for the sanitation infrastructure, wastewater is discharged directly into the ravines through small tubes [8].

The agricultural town E. Portes Gil is located on a plain, with dominant Durisol soil, and subhumid temperate climate with summer rains. The temperature ranges between 14–20 °C and precipitation between 800–1000 mm [48].

## 3. Materials and Methods

### 3.1. Sampling and Analytical Methods

In 2016, water samples from 15 wells and a spring were randomly collected in EPG. Sampling was carried out in June, August, October (rainy season), and in May and November (dry season). The methodology used for the water analyses was based on standard American Public Health Association—American Water Works Association (APHA–AWWA) methods [49]. In situ, the water temperature, pH, and dissolved oxygen (Winkler method) were measured in triplicate, with a portable meter Orion brand name No. 9107 water proof pH triode, and a dissolved oxygen Test Kit HANNA® HI 3810 (110 test).

Water samples (500 mL) were collected in triplicate for the analysis of 9 heavy metals: Al, Fe, Cu, Pb, Cd, Zn, Co, Ni, and Cr. For the determination of dissolved metals the samples were filtered through a 0.45 micron pore membrane (Wathman), and later they were preserved by adding concentrated nitric acid until obtaining a pH < 2. All the samples were refrigerated at 4 °C and were analyzed within 30 days of the collection.

The concentration of these elements was determined using a double flame atomic absorption spectrophotometer, varian 55B ray and hydride generator (VGA 77), and a coupled digester oven (Mars press), as indicated by NMX-AA-051-SCFI-2001 [50]. The standards of the metal solutions used for the Atomic Absorption were from the Accutrace TM Reference Standard.

A household survey (110 total) was applied to obtain information on water supply and sanitary measures. The Kruskal-Wallis test was applied to the results obtained, using the Statgraphics Centuriun software. The values obtained were compared with the Official Mexican Standards (NOM) CE-CCA-001/89 [51], and NOM-127-SSA1-94-Modification (2000) [52] in order to know the quality of the water that is taken in.

### 3.2. Evaluation of the Risk of Diseases due to the Consumption of Metals in Groundwater

For the estimation of the health risk, chronic exposure throughout the life of a specific metal was estimated by means of the chronic daily intake (CDI) equation (USEPA method) [38,39]:

$$\text{CDI (mg/kg/d)} = C \times \text{DI/BW}, \tag{1}$$

where:

C = Concentration of the metal in the drinking water (mg/L)
DI = Average daily drinking water intake (L/d)
BW = Body weight (kg)

The average weights of each subgroup of the population: Adults, adolescents, and children were obtained from the National Institute of Public Health, as well as the average daily drinking water intake of each subgroup [53,54].

### 3.2.1. Risk of No Cancer Effect

The hazard quotient (HQ) equation (Equation (1)) was used to estimate the risk of noncancerous diseases, due to the intake of a particular heavy metal in well water and spring water [38,39].

$$\text{HQ}_{\text{Metal}} = \text{CDI/RfD (dimensionless)}, \tag{2}$$

where CDI is the chronic daily intake and RfD is the reference dose of the metal by oral exposure (threshold dose). The values of the RfD (threshold dose in mg/kg/d) were obtained from the U.S.

EPA–IRIS (for its acronym in English, integrated risk information system) [55,56]. The HQ was calculated for each of the metals and for the population subgroups.

The establishment of health risk caused by metals was interpreted based on the values of HQ. Values lower than 1 mean that there is no risk, and greater than 1 mean that there is a probability of an adverse effect on health [38,39,41,53].

The potential risk of noncarcinogenic effects from the combined effect (all heavy metals) of chronic exposure via ingestion was calculated using the hazard index (HI), which is the sum of the HQs [38]:

$$\text{HI (chronic effect)} = \text{HQ}_{\text{metal 1}} + \text{HQ}_{\text{metal 2}} \cdots + \text{HQ}_{\text{metal n}}. \tag{3}$$

### 3.2.2. Risk for Cancer Effect

The risk of cancer associated with oral exposure (ingestion) was calculated using the following equation [38,41,42]:

$$\text{CRI} = \text{CDI} \times \text{SF (dimensionless)}, \tag{4}$$

where CRI is the probability of developing cancer at some point in life as a result of exposure to a potential carcinogen (Pb and Cr in the study), CDI the chronic daily intake, and SF the carcinogenic potency slope factor.

The risk of cancer from the exposure of the group of potentially carcinogenic metals was calculated by adding the RICs of each metal to the subgroup of the population [38,39].

$$\text{CRI}_{\text{total}} = \text{CRI}_{\text{metal 1}} + \text{CRI}_{\text{metal 2}} + \text{CRI}_{\text{metal n}} \rightarrow. \tag{5}$$

The generally acceptable limit of the individual carcinogenic risk increase is $1 \times 10^{-6}$, that is, the probability that an individual will develop cancer for every million people. However, for the risk corresponding to more than one substance (mixtures) in polluted sites, exposure levels that result in an increase in cancer between $10^{-4}$ and $10^{-6}$ are considered acceptable and are generally used as a reference point $1 \times 10^{-5}$ [38,39]. Age-dependent adjustment factors (ADAF) were applied to the values of the slopes of carcinogenic power (SF) for water, collected in the database of U.S. EPA–IRIS, according to the following criteria:

For children < 2 years old: Value of SF $\times$ 10
For children 2–16 years: Value of SF $\times$ 3
For adolescents > 16 years old: SF value $\times$ 1.

## 4. Results

### 4.1. Water Quality in the Population of EPG

In the wells and in the spring, the presence of seven heavy metals of the nine analyzed in the water was detected (Tables 1 and 2). In both types of water sources, the tendency of the concentration of metals to increase during the dry season was observed. Concentrations higher than 200% were registered in relation to rainfall measurements in well water; which were statistically significant for the case of Fe, Zn, Ni, Pb, and Cu. When comparing the concentrations obtained with the quality standards for human use and consumption, it was observed that the concentrations of Fe, Ni, and Al in the wells exceeded the maximum permissible limits (MPL) based on the Official Mexican Standard (NOM based on its initials in Spanish). The Pb registered slightly higher average values (0.012 mg/L) than the MPL, according to NOM-127-Modified in 2000 (MPL = 0.01 mg/L); in some of the samples, concentrations were estimated up to 0.052 mg/L, that is, five times higher than the MPL. The concentration of the rest of the metals (Zn, Cu, Cr) remained below the MPL. With respect to the spring, the same pattern of increase in the dry season was observed for all the detected metals, which was significant for Cu, Fe, and Zn. The Fe in the spring water registered higher concentrations than the MPL; same for the Pb, particularly in the dry season, according to NOM-127.

**Table 1.** Heavy metals in well water. Descriptive statistics, significance value of the Kruskal-Wallis Test for the comparison between rain and drought, and critical values according to two Mexican standards.

| Parameters (n = 45) | Well Mean | SD * | Min. | Max. | Rainy (mean) | Dry (mean) | *p*-Value | NOM-127 [+] | CE |
|---|---|---|---|---|---|---|---|---|---|
| T (°C) | 21.17 | 1.2 | 20 | 25 | 20.70 | 22.56 | 0.001 * | - | - |
| pH | 7.38 | 0.2 | 6.8 | 7.8 | 7.30 | 7.6 | 0.000 * | 6.5–8.5 | 5–9 |
| $O_2$ (mg/L) | 3.09 | 0.8 | 1.6 | 4.5 | 2.98 | 3.42 | 0.181 | - | 4 |
| Al (mg/L) | 0.062 | 0.040 | 0.00 | 0.156 | 0.016 | 0.108 | 0.150 | 0.20 | 0.02 |
| Fe (mg/L) | 0.394 | 0.424 | 0.00 | 1.900 | 0.107 | 0.682 | 0.010 * | 0.3 | 0.3 |
| Zn (mg/L) | 0.018 | 0.022 | 0.00 | 0.111 | 0.004 | 0.033 | 0.009 * | 5 | 5 |
| Ni (mg/L) | 0.013 | 0.026 | 0.00 | 0.089 | 0.003 | 0.024 | 0.019 * | - | 0.01 |
| Pb (mg/L) | 0.006 | 0.015 | 0.00 | 0.056 | 0.000 | 0.012 | 0.000 * | 0.01 | 0.05 |
| Cr (mg/L) | 0.001 | 0.002 | 0.00 | 0.011 | 0.001 | 0.002 | 0.721 | 0.05 | 0.05 |
| Cu (mg/L) | 0.015 | 0.018 | 0.00 | 0.089 | 0.001 | 0.030 | 0.000 * | 2 | 1 |

\* SD: Standard Deviation. [+] NOM-127 (Modified in DOF 2000).

**Table 2.** Heavy metals in the spring water. Descriptive statistics, significance value of the Kruskal-Wallis Test for the comparison between rain and drought, and critical values according to two Mexican standards.

| Parameters (n = 15) | Spring Mean | SD * | Min. | Max. | Rainy (mean) | Dry (mean) | *p*-Value | NOM-127 [+] | CE |
|---|---|---|---|---|---|---|---|---|---|
| T (°C) | 21.75 | 0.866 | 21 | 23 | 21.33 | 23 | 0.006 * | - | - |
| pH | 7.24 | 0.215 | 6.8 | 7.4 | 7.20 | 7.37 | 0.240 | 6.5–8.5 | 5–9 |
| $O_2$ (mg/L) | 1.96 | 0.198 | 1.6 | 2.3 | 2.00 | 1.83 | 0.108 | - | 4 |
| Al (mg/L) | 0.007 | 0.013 | 0.00 | 0.044 | 0.000 | 0.014 | 0.102 | 0.20 | 0.02 |
| Fe (mg/L) | 0.209 | 0.208 | 0.00 | 0.533 | 0.000 | 0.418 | 0.001 * | 0.3 | 0.3 |
| Zn (mg/L) | 0.011 | 0.012 | 0.00 | 0.033 | 0.000 | 0.022 | 0.001 * | 5 | 5 |
| Ni (mg/L) | 0.001 | 0.003 | 0.00 | 0.011 | 0.000 | 0.003 | 0.102 | - | 0.01 |
| Pb (mg/L) | 0.008 | 0.007 | 0.00 | 0.020 | 0.003 | 0.013 | 0.142 | 0.01 | 0.05 |
| Cr (mg/L) | 0.006 | 0.006 | 0.00 | 0.033 | 0.002 | 0.011 | 0.363 | 0.05 | 0.05 |
| Cu (mg/L) | 0.018 | 0.018 | 0.00 | 0.044 | 0.000 | 0.036 | 0.001 * | 2 | 1 |

\* SD: Standard Deviation, [+] NOM-127 (Modified in DOF 2000).

### 4.2. Health Risk Assessment

### 4.2.1. Risk of Effect No Cancer

The risk of noncancerous diseases due to the chronic ingestion of heavy metals present in spring water and wells is greater for children than for adults (Tables 3 and 4, Figure 2). This is evidenced by the values of the hazard quotients (HQ), as well as in the hazard indices (HI).

When comparing the health risk associated with the oral exposure of the pool of metals (HI) from the wells and the spring, it was observed that the risk is greater, due to the intake of water from the spring, mainly due to the relatively high value of the quotient of danger for chromium (HQ). However, in none of the cases is the risk considered high, since the HQs were less than 1.

By contrasting the HI (index expressing the risk by the combined effect of metals) between the study seasons, it was observed that this was notably higher in the dry season, both due to the consumption of water from wells and from the spring, detecting a greater difference between the magnitude of the risk between dry and rainy seasons for the subgroup of children compared to the subgroup of adults; this suggests that although the risk is higher in the dry season for the three subgroups evaluated, it is even more acute for children.

**Table 3.** Hazard quotient (HQ) and hazard Index (HI), for the water intake of the wells, in the subpopulation of children, adolescents and adults of E. Portes Gil.

| Heavy Metals | Well Water (Mean) | | | Well Water Rainy (Mean) | | | Well Water Dry (Mean) | | |
|---|---|---|---|---|---|---|---|---|---|
| HQ | Children | Teens | Adults | Children | Teens | Adults | Children | Teens | Adults |
| Al | 0.009 | 0.005 | 0.004 | 0.002 | 0.001 | 0.001 | 0.016 | 0.009 | 0.008 |
| Fe | 0.012 | 0.007 | 0.005 | 0.003 | 0.002 | 0.001 | 0.020 | 0.012 | 0.010 |
| Zn | 0.001 | 0.001 | 0.001 | 0.000 | 0.000 | 0.000 | 0.002 | 0.001 | 0.001 |
| Ni | 0.014 | 0.008 | 0.007 | 0.003 | 0.002 | 0.001 | 0.025 | 0.015 | 0.012 |
| Pb | 0.003 | 0.002 | 0.002 | 0.000 | 0.000 | 0.000 | 0.007 | 0.004 | 0.003 |
| Cr | 0.010 | 0.006 | 0.005 | 0.007 | 0.004 | 0.003 | 0.014 | 0.008 | 0.007 |
| Cu | 0.009 | 0.005 | 0.004 | 0.001 | 0.000 | 0.000 | 0.017 | 0.010 | 0.008 |
| **HI** | **0.059** | **0.035** | **0.028** | **0.017** | **0.010** | **0.008** | **0.102** | **0.060** | **0.047** |

**Table 4.** HQ and HI, for the water intake of the spring water, in the subpopulation of children, teenagers and adults of E. Portes Gil.

| Heavy Metals | Spring Water (Mean) | | | Spring Water Rainy (Mean) | | | Spring Water Dry (Mean) | | |
|---|---|---|---|---|---|---|---|---|---|
| HQ | Children | Teens | Adults | Children | Teens | Adults | Children | Teens | Adults |
| Al | 0.001 | 0.001 | 0.000 | 0.000 | 0.000 | 0.000 | 0.002 | 0.001 | 0.001 |
| Fe | 0.006 | 0.004 | 0.003 | 0.000 | 0.000 | 0.000 | 0.012 | 0.007 | 0.006 |
| Zn | 0.001 | 0.000 | 0.000 | 0.000 | 0.000 | 0.000 | 0.002 | 0.001 | 0.001 |
| Ni | 0.002 | 0.001 | 0.001 | 0.000 | 0.000 | 0.000 | 0.003 | 0.002 | 0.001 |
| Pb | 0.005 | 0.003 | 0.002 | 0.002 | 0.001 | 0.001 | 0.008 | 0.004 | 0.004 |
| Cr | 0.045 | 0.027 | 0.021 | 0.014 | 0.008 | 0.007 | 0.077 | 0.045 | 0.036 |
| Cu | 0.010 | 0.006 | 0.005 | 0.000 | 0.000 | 0.000 | 0.020 | 0.012 | 0.009 |
| **HI** | **0.070** | **0.041** | **0.033** | **0.016** | **0.009** | **0.007** | **0.124** | **0.073** | **0.058** |

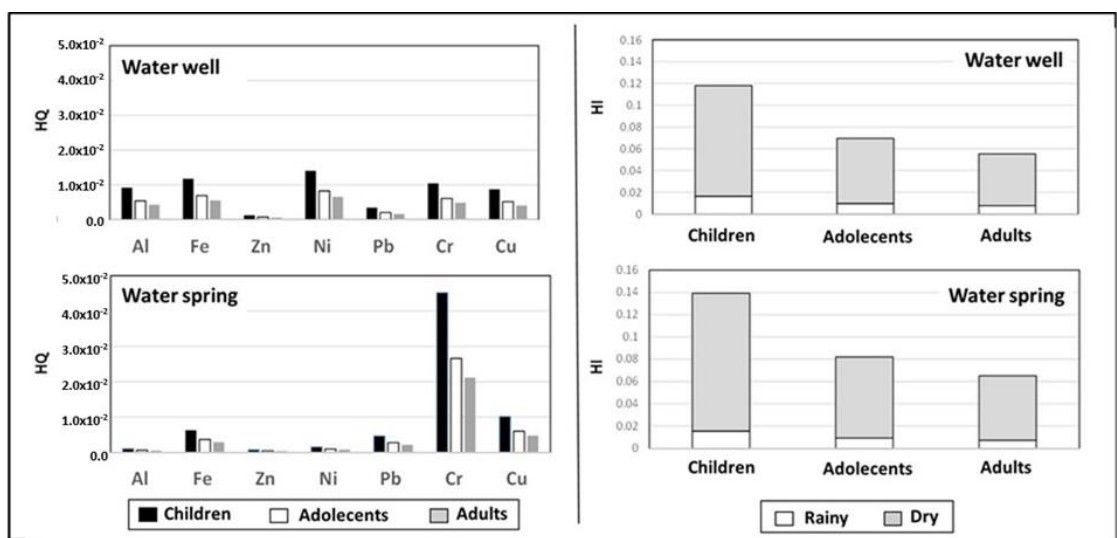

**Figure 2.** HQ and HI, due to the intake of water from the wells and the spring, considering the different population subgroups (children, teenagers and adults) and seasons (rain and dry).

### 4.2.2. Risk for Cancer Effect

Table 5 and Figure 3 show that the risk of carcinogenic diseases is greater due to the prolonged intake of water from the spring, compared to the water in the wells, particularly for children, which is explained by the presence of higher concentrations of lead and chromium in this source of water. These risks are accentuated in the dry season, mainly in the subgroup of children. The CRI (cancer risk index) obtained for the subgroup of children with spring water showed a value that can be considered critical.

**Table 5.** Cancer risk index considering the individual and total effect of the potentially carcinogenic metals present at the spring W and the wells W., for the subpopulation of children, teenagers and adults of E. Portes Gil.

| Heavy Metals | Children | | Teenagerss | | Adults | |
|:---:|:---:|:---:|:---:|:---:|:---:|:---:|
| **CRI** | **Spring W** | **Well W** | **Spring W** | **Well W** | **Spring W** | **Well W** |
| **Chrome** | $3.2 \times 10^{-4}$ | $7.5 \times 10^{-5}$ | $3.9 \times 10^{-5}$ | $9.2 \times 10^{-6}$ | $3.1 \times 10^{-5}$ | $7.3 \times 10^{-6}$ |
| **Lead** | $6.8 \times 10^{-6}$ | $5.1 \times 10^{-6}$ | $8.0 \times 10^{-7}$ | $6.0 \times 10^{-7}$ | $7.0 \times 10^{-7}$ | $5.0 \times 10^{-7}$ |
| **$\sum$CRI** | $\mathbf{3.3 \times 10^{-4}}$ | $\mathbf{8.0 \times 10^{-5}}$ | $\mathbf{4.0 \times 10^{-5}}$ | $\mathbf{9.8 \times 10^{-6}}$ | $\mathbf{3.2 \times 10^{-5}}$ | $\mathbf{7.8 \times 10^{-6}}$ |

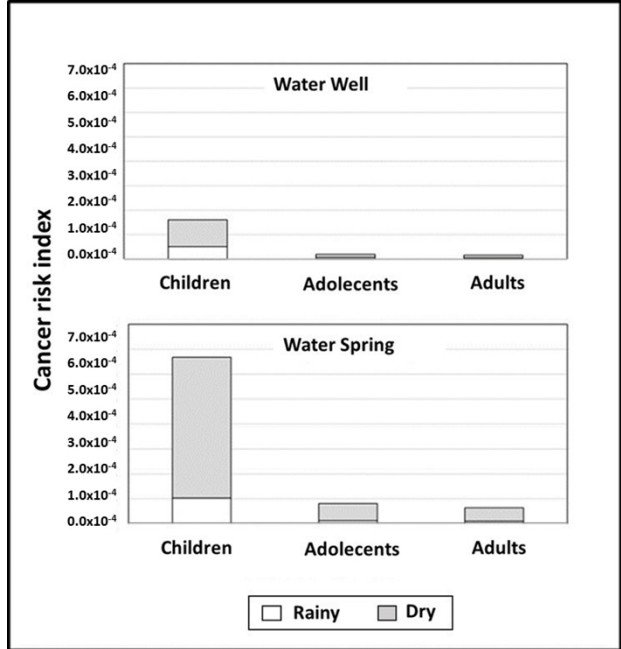

**Figure 3.** Total cancer risk index due to the intake of spring water metals and wells water, for the subpopulation of children, adolescents and adults of E. Portes Gil.

## 5. Discussion

The population of EPG is more susceptible to disease from the intake of spring water than well water, due to the presence of higher concentrations of chromium and lead, as both metals are toxic to human health according to the IARC (International Agency for Cancer Research, 2015) [37]. Many people in the village are exposed to these pollutants, since all homes (522 inhabitants) receive water from this source through the pipeline and use it for drinking, cooking, washing, etc. On the other hand, wells are present in 70% of homes, and only in 41% are they used for drinking [8].

In the analysis, the potential health risk due to prolonged exposure to chromium was evident, with greater values of cancer risk indices (CRI) than the acceptable limit of $1 \times 10^{-6}$ [38], especially for the groups of children and teenagers during the dry season. Although chromium forms part of the earth's crust and can be released to the environment by natural sources, its release is mainly due to anthropogenic processes [57–60]. Chromium is widely used in industries, such as metallurgy, electroplating, paint and pigments, textile production, leather tanning, wood preservation, and paper chemical production, among others [22,58–61]. These industries are responsible for the high concentration of chromium in the environment, since they release their wastewater to the rivers without proper treatment [22,58].

In the metropolitan area of Puebla-Tlaxcala (MAPT), there are hundreds of industries using different processes, with an important proportion of those using chromium compounds for production or manufacturing processes, such as textiles, metalworking, paints, automotive [8,18,21], etc. These,

like the rest of the factories, do not treat their wastewater in the proper way to remove heavy metals and other chemical pollutants [7,8]. Thus, in this river, the waste of the fourth largest metropolitan area of Mexico (ZMPT) is discharged, and at the same time, agricultural populations like EPG, located in the lower areas of the basin, use the polluted water of the river for the irrigation of their crops [8].

The release of chromium into the environment is also related to modern agricultural practices, fertilizers, and irrigation using wastewater [58]. Both activities are carried out intensively in the study area, because, in addition to agrochemicals, it is irrigated exclusively with polluted water from the Atoyac River. This may explain the high concentrations of chromium in the soils of EPG reported in 2016 by Ávila González [62], whose values were three times higher than the permissible limit for agricultural soils, based on the standard that establishes the criteria for which a polluted soil is determined, and other standards (NOM-147SEMARNAT/SSA1-2004) [63]. In the research developed by Castro-González et al. (2017) about the risks of exposure to heavy metals in populations located in the Alto Atoyac basin [42], chromium in soils was reported as the metal responsible for the risk of diseases in men, due to chronic dermal exposure, since every day of the year, their skin (hands, face, feet) are in contact with the soil by the activities of tillage and harvest, among others, which they carry out without protective equipment [42].

The literature mentions that the pollution of soils by the retention of chromium and other heavy metals is due to the use of wastewater and fertilizers and can contaminate groundwater [5,9,11,64,65]. In EPG, this represents a possibility, because polluted water from the Atoyac River flows through dozens of irrigation canals around the village and the wells are shallow (10–20 m) [8] and have been constructed without the advice of experts, so they are more vulnerable to pollution.

The health effects of chromium compounds may vary with the route of exposure. For example, respiratory effects are associated with the inhalation of chromium compounds, but not with oral and dermal exposures, and gastrointestinal effects are mainly associated with oral exposure. In general, the main white organs for the toxic action of chromium are bronchi, gastrointestinal tract, liver, and kidney [5,36,60]. The effects of exposure to any hazardous substance depend on the dose, duration, how they are exposed, personal traits and habits, and the presence or absence of other chemicals [22,41,66].

In addition to the presence of chromium in groundwater, lead was also detected, with concentrations greater than the maximum permissible limit for human consumption in some well samples. Lead, unlike chromium and the other metals analyzed, is not required for the human metabolism, and therefore is dangerous when entering the human body orally, even in low amounts [28,41]. Lead and its alloys are common components of pipes, batteries, cable linings, etc., and the compounds of this metal are used as pigments in paints, in varnishes for ceramics, and in filling materials [22,28,67]. Lead is found naturally in the environment. However, most of the high levels found in the environment originate from human activities. Environmental levels of lead have increased more than a thousand times during the last three centuries as a result of human activity [28].

Lead is found in wastewater from industries that handle lead (mainly the iron and steel industries and those that manufacture lead), runoff water in urban centers, and mineral piles [36,60,61]. People can be exposed to lead by eating food or drinking water containing lead [36,67]. Lead mainly affects the nervous system, both in children and adults [68]. The EPA and the International Agency for Research on Cancer (IARC) have determined that inorganic lead is probably carcinogenic in humans. Children are more vulnerable than adults to lead poisoning, and it affects them in different ways, depending on how much lead a child ingests. A child who ingests large amounts of lead can develop anemia, kidney damage, colic (severe stomach pain), muscle weakness, brain damage, and may eventually die. At even lower exposure levels, lead can affect a child's physical and mental development [22,25,28,36,68–70]. In the case of girls, the situation may be aggravated because they develop physiologically earlier than boys, due to the early release of estradiol, which triggers the growth and maturation of bones, together with a greater accumulation of fatty tissue [41,66]. It is also important to mention that the women of the village spend more time inside their houses and their exposure to spring water in relation to men is

greater. For this reason, this situation could explain the higher incidence of kidney and liver diseases in the female group in the town.

An important aspect to consider is that the estimated indices only provide information on the magnitude of the risk associated with oral exposure to heavy metals contained in drinking water.

This implies that although for most heavy metals, HQ < 1 means safe, it must be considered that exposure to heavy metal can also occur, through other exposure routes, such as intake of contaminated plants, since most plants can absorb and bioaccumulate soil pollutants [45,71]. Consequently, human health is directly affected by the consumption of these plants, since there is no effective mechanism of excretion [44]. The study by Latif et al. (2018) [72] shows how vegetables irrigated with wastewater differ in their potential to accumulate metals, and have Cr concentrations above the limit allowed by FAO in spinach, pumpkins, eggplants, etc. In the town of E Portes Gil, several plant species are cultivated for human consumption, such as pumpkins, parsley, coriander, among others, so that the risks to which the inhabitants of this agricultural town are exposed, are probably greater than those estimated. The research carried out by Li (2018a and b) [73,74]; show that human exposure to pesticides occurs through different routes, therefore, it is necessary to analyze the factors of total risk to health, considering the calculated individual factors, such as exposure to soil, drinking water and agricultural foods. Based on these considerations and our results, it is necessary that in the agricultural locality E. Portes Gil, environmental remediation measures be implemented, since the risks to human health could be greater when considering other possible routes of exposure.

## 6. Conclusions

The results of the research revealed that the chronic consumption of groundwater, particularly that coming from the spring (the main source of supply), represents a risk of developing carcinogenic diseases, particularly in children, due to the presence of metals, such as chromium and lead. A pattern of increase in the concentration of these and other heavy metals in the water of the wells and the spring is detected in the dry season, in which there is a continuous flow of polluted water from the Atoyac River for agricultural use, which suggests pollution of these sources of supply by possible infiltration. It is necessary to improve the control of these groundwater sources in terms of their infrastructure and protection to avoid the possible pollution of the resource, as well as the application of some treatment measures, taking into account that in the locality, only the residual water of the Atoyac river is used for the irrigation of crops and that the population is exposed by different routes.

**Author Contributions:** A.H.S. conceived the project and designed the experiments; G.P.C. performed the experiments; E.C.R. performed the heavy metal analysis in the water, J.L.M.P. and W.A.G.S. analyzed the data; A.C.M. sample analysis and fieldwork; all authors contributed with reagents, materials, and equipment; G.P.C. and A.H.S. wrote the paper.

**Funding:** Thanks to Consejo Nacional de Ciencia y Tecnología (CONACYT) for the Gabriela Pérez Castresana scholarship (601562), and Vicerectoria de Investigación y Estudios de Posgrado (VIEP) of the Benemérita Universidad Autónoma de Puebla (BUAP) for the support of this research work.

**Conflicts of Interest:** The authors have no conflict of interest to declare.

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
