# Peer review of "Evaluation of Health Risks Due to Heavy Metals in a Rural Population Exposed to Atoyac River Pollution in Puebla, Mexico"

_water, doi:10.3390/w11020277_

Round 1
Reviewer 1 Report
The authors conducted a very important study titled "Evaluation of Health Risks in a Rural Population Exposed to Atoyac River Pollution in Puebla, Mexico.", which is good for researchers in the field of environmental health. The whole manuscript is well written and organized. The topic is important and highly related to the journal. It is suggested that this manuscript should be accepted with some minor revisions.
The significance of the study should be addressed more in the introduction part.
What is the human health management implementations of the results?
Human health risk assessment: for most heavy metals, the HQ<1 means safe, and this must be further discussed. Please note that exposure to the heavy metal via water is only one of the major human exposure pathways of the contaminants. Other notable exposure pathways of the contaminants could occur via the ingestion of contaminated plants (like medical plants, tea leaves, and other agricultural commodities) since most plants can uptake and bioaccumulate soil contaminants. Thus, unless the human health risk assessment is based on all possible exposures, the computed HQ values might not be low enough to allow for additional exposures and thus may not be sufficiently protective of human health. (recommended references about the comprehensive human exposure study:
https://doi.org/10.1016/j.envint.2018.10.047
https://doi.org/10.1016/j.scitotenv.2018.05.148
It means that local official must take remediation actions regarding the heavy metal risk assessment efforts and results because human health risks could increase with consideration of other possible exposure pathways.
Author Response
Point-by-point response to reviewer 1
First, we would like to express our gratitude for the valuable and constructive comments to our manuscript. yours suggestions were incorporated to the manuscript and now appear in the second version.
General comments:
1. The significance of the study should be addressed more in the introduction part. R. Thank you. We have written in detail the importance of the study in the introduction lines 82-93.
2. What is the human health management implementations of the results? R. This information is detailed in the five paragraph of introduction lines 83-94 and in the last paragraph of the discussion lines 317-333.
3. Human health risk assessment: for most heavy metals, the HQ<1 means safe, and this must be further discussed. Please note that exposure to the heavy metal via water is only one of the major human exposure pathways of the contaminants. Other notable exposure pathways of the contaminants could occur via the ingestion of contaminated plants (like medical plants, tea leaves, and other agricultural commodities) since most plants can uptake and bioaccumulate soil contaminants. Thus, unless the human health risk assessment is based on all possible exposures, the computed HQ values might not be low enough to allow for additional exposures and thus may not be sufficiently protective of human health. (recommended references about the comprehensive human exposure study: https://doi.org/10.1016/j.envint.2018.10.047 https://doi.org/10.1016/j.scitotenv.2018.05.148
It means that local official must take remediation actions regarding the heavy metal risk assessment efforts and results because human health risks could increase with consideration of other possible exposure pathways. R. Thanks, we review the recommended references, they were very useful to argue that the risks to human health could be greater when considering other possible routes of exposure to heavy metals. Lines 328-333.

Reviewer 2 Report
Dear author,
the paper titled "Evaluation of Health Risks in a Rural Population Exposed to Atoyac River Pollution in Puebla, Mexico" can be published in the journal "Water" since include evaluation of health risks in rural population as well as include ecological risks of heavy metal contamination. Some minor revision is needed. Title must include one of the main objective of this work, namely determination of nine heavy metals (please see lines 17/18 of the abstract). More over sub chapter 3.1. must include more information about the efficiency of the analytical procedures of the heavy metals. Authors should provide if possible, the data on OA/QC. Did they use water standard reference material (or some other method) to monitor the accuracy of the analytical procedure? Authors should point this in the manuscript.
Author Response
Point-by-point response to reviewer 2
First, we would like to express our gratitude for the valuable and constructive comments to our manuscript. yours suggestions were incorporated to the manuscript and now appear in the second version.
General comments:
1. Title must include one of the main objective of this work, namely determination of nine heavy metals (please see lines 17/18 of the abstract). R. Thanks for the suggestion, we changed the title in order to include the research objectives. The title modified is: Evaluation of Health Risks due to Heavy Metals in a Rural Population Exposed to Atoyac River Pollution in Puebla, Mexico (lines 2-4)
2. More over sub chapter 3.1. must include more information about the efficiency of the analytical procedures of the heavy metals. Authors should provide if possible, the data on OA/QC. Did they use water standard reference material (or some other method) to monitor the accuracy of the analytical procedure? Authors should point this in the manuscript. R. We have included in sub chapter 3.1 the requested information about the heavy metal analytical procedure and the use of standards. Lines 125-136.

Reviewer 3 Report
The topic falls in the scope of “Water” journal, and the study provides meaningful information especially in the context of the studied area. Experiments seem well designed and methods used appropriated. The objectives of the work were met. The set of data presented is enough and literature was critically explored to support the authors' hypotheses. However, there are still some issues that have to be addressed by the authors before considering the manuscript for publication. My major concern is related to the concept of “risk”. I also consider that the research is too regional in content; Therefore, the data may have regional interest. My comments are detailed below.
Some minor typos, grammar and syntax errors should be carefully revised and corrected accordingly. For example, in line 3 (Title) “Rive” should be “River”
Abstract
I suggest the authors to include some quantitative data which could be more interesting and informative for the readers.
Keywords
Authors should rephrase keywords. Do not use words or terms in the title as keywords: the function of keywords is to supplement the information given in the title. Words in the title are automatically included in indexes, and keywords serve as additional pointers.
2. Study area
Fig.1: The location on the map of México should be provided. In addition, all maps (and satellite images) must indicate scale and orientation.
This sub-section (Study area) needs more information. Please provide information on climatic parameters, namely precipitation and temperature, geology, soil types, vegetation, etc.
3. Materials and Methods
In this section you should specify the characteristics of all equipments (report model, brand name, city and country of manufacturer). For example, the equipment for in situ measurements is not mentioned.
Lines 105-113: How much time passed between water sampling in the field and the analysis? Water samples were filtered in the field or in the lab? Was pH-adjusted in the field to minimize the possibility of metal loss?
This section should also give more details about Quality Assurance and Quality Control. Authors should indicate the obtained accuracy values. Were used reference materials? If so, these should be listed. It would be also interesting to provide the reader with limits of detection/determination of analyzed elements.
The “chronic daily water intake (CDI)”, “hazard quotient (HQ)” and “hazard index (HI)” are of interest, but chemistry is generally only part of the triad used to assess risk in water. No biology is invoked here. Chemical levels are not risk assessments – they are hazard assessments – risk is a function of hazard and exposure. While looking at total concentration elements is part of that overall assessment, the authors do not really engage in what their data mean in terms of the concepts of risk (and risk management). Throughout the manuscript, authors should review the use of the term “risk”.
In addition, in this study, the authors only analyzed water samples. All other parameters, necessary for the calculations, are taken from the bibliography. Therefore, they are merely indicative and no conclusions can be drawn about this study area.
Author Response
Point-by-point response to reviewer 3
First, we would like to express our gratitude for the valuable and constructive comments to our manuscript. Yours suggestions were incorporated to the manuscript and now appear in the second version.
General comments:
1. Some minor typos, grammar and syntax errors should be carefully revised and corrected accordingly. For example, in line 3 (Title) “Rive” should be “River”
R. Thank you. Misspellings in title and text have been corrected.
2. Abstract. I suggest the authors to include some quantitative data which could be more interesting and informative for the readers.
R. We have included quantitative data in the abstract in lines 23, 25 and 29.
3. Keywords. Authors should rephrase keywords. Do not use words or terms in the title as keywords: the function of keywords is to supplement the information given in the title. Words in the title are automatically included in indexes, and keywords serve as additional pointers.
R. Thank you, the keywords were rephrased in order to include new words not included in the title in line 30.
4. Study area. Fig.1: The location on the map of México should be provided. In addition, all maps (and satellite images) must indicate scale and orientation.
R. We have included the map of Mexico in figure 1, this figure have scale and orientation as you suggest.
5. This sub-section (Study area) needs more information. Please provide information on climatic parameters, namely precipitation and temperature, geology, soil types, vegetation, etc.
R. We have described the climatic parameters in lines 116-118.
6. 3. Materials and Methods. In this section you should specify the characteristics of all equipments (report model, brand name, city and country of manufacturer). For example, the equipment for in situ measurements is not mentioned.
R. We have described more detailly the methods including the equipment characteristics. Lines 125-136.
7. Lines 105-113: How much time passed between water sampling in the field and the analysis? Water samples were filtered in the field or in the lab? Was pH-adjusted in the field to minimize the possibility of metal loss?
R. We have included the requested parameters in lines 128-132.
8. This section should also give more details about Quality Assurance and Quality Control. Authors should indicate the obtained accuracy values. Were used reference materials? If so, these should be listed. It would be also interesting to provide the reader with limits of detection/determination of analyzed elements.
R. We have described the characteristics of standards as you suggest us in lines 135-136.
9. The “chronic daily water intake (CDI)”, “hazard quotient (HQ)” and “hazard index (HI)” are of interest, but chemistry is generally only part of the triad used to assess risk in water. No biology is invoked here. Chemical levels are not risk assessments – they are hazard assessments – risk is a function of hazard and exposure. While looking at total concentration elements is part of that overall assessment, the authors do not really engage in what their data mean in terms of the concepts of risk (and risk management). Throughout the manuscript, authors should review the use of the term “risk”.
R. This information is detailed in the five paragraph of introduction lines 82-93 and in the last paragraph of the discussion. lines 317-333.
10. In addition, in this study, the authors only analyzed water samples. All other parameters, necessary for the calculations, are taken from the bibliography. Therefore, they are merely indicative and no conclusions can be drawn about this study area.
R. We review new references, in order to argue that the risks to human health could be greater when considering other possible routes of exposure to heavy metals. Lines 328-333.

Round 2
Reviewer 3 Report
This is the second version of this manuscript, and it improves in both readability and scientific clarity with revision, so I commend the authors for their perseverance. However, I insist on the correction of an issue that I had already mentioned in my previous review (detailed below).
As the main purpose of this paper was “...to determine the concentration of nine heavy metals in the wells and a spring…”, the authors are not really evaluating “risk” but rather “hazard”. Chemical levels are not risk assessments – they are hazard assessments – risk is a function of hazard and exposure. No biology is invoked here. While looking at total concentration elements is part of that overall assessment, the authors do not really engage in what their data mean in terms of the concepts of risk (and risk management). The authors seem to be reluctant to substitute the term “risk” for “hazard, which would make terminology more adequate and accurate. It is an option of the authors, which I respect. But that I cannot consider scientifically correct in order to consider the manuscript suitable for publication.
Author Response
Dear Professor:
We received a letter dated January 21, in which you suggests us make minor revisions to our manuscript entitled “Evaluation of Health Risks in a Rural Population Exposed to Atoyac”, submitted to the journal Water.
Point-by-point response to reviewers´ criticisms
We have carefully revised your comments and recommendations. We would like to express our sincere gratefulness for yours suggestions, however we cannot substitute the term “risk” for “hazard, because the present research is based on the 1986 Guidelines for the Health Risk Assessment of Chemical Mixtures proposed by the United States Environmental Protection Agency (USEPA), a procedural guide for evaluating data on the health risks from exposures to chemical mixtures. Cited in the reference 39, lines 413-416.
It is important to highlight that the principles and concepts put forth in the Guidelines remain in effect. However, in 2000, EPA issued a new document, Supplementary Guidance for Conducting Health Risk Assessment of Chemical Mixtures (EPA/630/R-00/002 - August 2000). While the Guidelines describe broad principles and include few specific procedures, the 2000 guidance is a supplement that is intended to provide more detail on these principles and their applications.
The principles and concepts put forth in the Guidelines cannot be altered. We cannot substitute the term “risk” for “hazard” as you suggests.
We hope this revised version be suitable for its publication. Thank you very much in advance.
Hoping to hearing you.
Anabella Handal Silva, PhD.
Departamento de Biología y Toxicología de la Reproducción
Instituto de Ciencias, benemérita Universidad Autónoma de Puebla.
Av. 14 Sur No. 6301 Ciudad Universitaria, Puebla, Pue, México. CP. 72570
Tel. (+52) 222 229 5500 (Ext. 7160, 7161)
